# Association between Anemia Severity and Ischemic Stroke Incidence: A Retrospective Cohort Study

**DOI:** 10.3390/ijerph20053849

**Published:** 2023-02-21

**Authors:** Hui-Fen Chen, Tsing-Fen Ho, Yu-Hung Kuo, Ju-Huei Chien

**Affiliations:** 1Department of Nephrology, Taichung Tzu-Chi Hospital, Buddhist Tzu-Chi Medical Foundation, No. 88 Fong-Shing Rd., Taichung City 42743, Taiwan; 2Department of Medical Laboratory Science and Biotechnology, Central Taiwan University of Science and Technology, No. 666 Buzih Rd., Taichung City 40601, Taiwan; 3Department of Research, Taichung Tzu-Chi Hospital, Buddhist Tzu-Chi Medical Foundation, No. 88 Fong-Shing Rd., Taichung City 42743, Taiwan

**Keywords:** anemia, ischemic stroke, elderly, hemoglobin, retrospective cohort study

## Abstract

Stroke patients presenting with anemia at the time of stroke onset had a higher risk of mortality and development of other cardiovascular diseases and comorbidities. The association between the severity of anemia and the risk of developing a stroke is still uncertain. This retrospective study aimed to evaluate the association between stroke incidence and anemia severity (by WHO criteria). A total of 71,787 patients were included, of whom 16,708 (23.27%) were identified as anemic and 55,079 patients were anemia-free. Female patients (62.98%) were more likely to have anemia than males (37.02%). The likelihood of having a stroke within eight years after anemia diagnosis was calculated using Cox proportional hazard regression. Patients with moderate anemia had a significant increase in stroke risk compared to the non-anemia group in univariate analyses (hazard ratios [HR] = 2.31, 95% confidence interval [CI], 1.97–2.71, *p* < 0.001) and in adjusted HRs (adj-HR = 1.20, 95% CI, 1.02–1.43, *p* = 0.032). The data reveal that patients with severe anemia received more anemia treatment, such as blood transfusion and nutritional supplementation, and maintaining blood homeostasis may be important to preventing stroke. Anemia is an important risk factor, but other risk factors, including diabetes and hyperlipidemia, also affect stroke development. There is a heightened awareness of anemia’s severity and the increasing risk of stroke development.

## 1. Introduction

Stroke is a leading cause of death and disability worldwide [1]. Stroke survivors suffer from various impairments and complications affecting motor, sensory, visual, language, and cognitive functions [2,3]. Therefore, a stroke imposes a great burden on patients as well as their caregivers and family members. Stroke patients may be hospitalized or may frequently visit the emergency department owing to their long-term sequelae and disability, which not only dramatically increases the burden on caregivers and their family’s finances, but also severely affects their quality of life. There are numerous recognized risk factors for stroke, such as hypertension, hyperlipidemia, diabetes mellitus, cigarette use, obesity, age, and physical activity [1,4,5]. Increases in the elderly population and life expectancy are also key reasons for the increase in number of stroke patients.

Anemia affects 15–32% of the world’s population, is usually present in stroke patients, and can worsen with aging [6,7]. In 2019, the age groups of 15 to 19 and 95 and older, for both males and females, had the highest global point prevalence of anemia. The mean (range) global prevalence rates of mild, moderate, and severe anemia were approximately 54.1%, (53.8–54.4%), 42.5% (42.2–42.7%), and 3.4% (3.3–3.5%), respectively [8]. Elderly individuals may experience malnutrition and dyspepsia as their physical condition deteriorates with age, and this may affect their hematopoiesis functions, thereby causing anemia or pancytopenia. Anemia is also a risk factor for ischemic stroke and is related to high post-stroke mortality [9,10].

Nevertheless, previous research has suggested that anemia may raise the risk of stroke. However, the new stroke guidelines from the American Stroke Association (ASA) do not list anemia as a major stroke risk factor [11]. Here, we conducted a retrospective cohort study to investigate the association between the severity of anemia and stroke incidence. Owing to Taiwan’s National Health Insurance (NHI) policy, anemia is rarely listed as a primary condition and may not be documented on patient medical records on the basis of *International Classification of Diseases, Tenth Revision (ICD-10)* codes. The laboratory data of anemia status were not available in Taiwan’s NHI system, and the prevalence of anemia could be underestimated. Moreover, the data of association between anemia and comorbidities in the Taiwanese population are scarce. An evaluation of the stroke risk factors, especially anemia severity, could provide important information that may enhance medical care or even national healthcare planning. This study retrospectively evaluated the prevalence and characteristics of anemia in hospitalized patients and analyzed whether anemia severity based on the hemoglobin (Hb) level was associated with stroke development.

## 2. Materials and Methods

### 2.1. Study Cohort

This retrospective cohort study included 454,424 patients aged ≥20 years who had visited or were hospitalized at Taichung Tzu-Chi Hospital, Taiwan, from 2013 to 2019. A total of 71,787 patients underwent at least 1 blood Hb measurement performed using a Sysmex XE-5000 hematology analyzer (Sysmex Co., Kobe, Japan) within 1 year to confirm their anemia status. This study was approved by the Research Ethics Committee of Taichung Tzu-Chi Hospital (REC 111-02). The need for informed consent was waived owing to the retrospective nature of the study and the use of anonymous medical records.

### 2.2. Definition of Anemia and ICD Codes

Adult patients older than the age of 20 were included in this study. All participants in this study completed at least one Hb measurement, and persons who did not fulfill the predetermined criteria were not included. The date of laboratory Hb measurement was defined as the index date, and the anemia severity was classified according to the World Health Organization (WHO) criteria [12]. We categorized the patients into different groups according to their anemia severity. Anemia is defined as an Hb level of <13.0 g/dL for men and <12.0 g/dL for women. The cutoff for Hb in mild anemia was 11.0–11.9 g/dL for women and 11.0–12.9 g/dL for men, whereas the cutoffs for moderate and severe anemia were 8.0–10.9 and <8.0 g/dL, respectively, for both men and women. As shown in Figure 1, the exclusion criteria were as follows: (1) patients without Hb measurements; (2) receiving a diagnosis that might affect the Hb status, including gastric intestinal bleeding (*ICD-10* code K92.2), bleeding (*ICD-10* code R58), trauma (*ICD-10* code T79.2), excessive bleeding associated with menopause onset (*ICD-10* code N92.4), intraoperative and postprocedural complications of spleen, endocrine, and nervous system (*ICD-10* code D78, E36, G97), excessive bleeding with onset of menstrual bleeding (*ICD-10* code N92.2), traumatic hemorrhage of the cerebrum (*ICD-10* code S06.360A), hemorrhage from respiratory passages (*ICD-10* code R04.9), nontraumatic intracerebral hemorrhage (*ICD-10* code I61.9), spleen diseases (*ICD-10* code D73), pulmonary vessels diseases (*ICD-10* code I28), stomach and duodenum diseases (*ICD-10* code K31), acute myocardial infarction (*ICD-10* code I21), injury to an unspecified body region (*ICD-10* code T14), or absent, scanty, or rare menstruation (*ICD-10* code N91), before their index date until anemia diagnosis; (3) receiving a stroke diagnosis before the index date on the basis of the *ICD-10* codes I63; (4) not visiting our out-patient clinic or being hospitalized within the last 2 years; and (5) death or leaving against medical advice (DAMA) less than 1 month after the index date.

A flowchart of the patient enrollment process is illustrated in Figure 1. All patients were grouped by sex and age (20–30, 31–40, 41–50, 51–60, 61–70, 71–80, and >80 years). The Hb status confirmation date was identified as the index date for the case and control groups, and stroke events were followed subsequently.

### 2.3. Outcome and Associated Factors

The eligibility of all patients was retrospectively determined in this cohort study. The severity of anemia was then subgrouped based on Hb level, and the stroke patients were those who had at least two *ICD-10* admission claims for clinic OPD visits or stroke-related hospitalization in our hospital during the study period. During the monitoring period, the occurrence of subsequent disease was examined. The occurrence of subsequent disease was analyzed during the observation period. Patients were individually tracked for 2–8 years, beginning on the index date, and followed thereafter. In this study, the outcome of stroke was defined as admission claims of *ICD-10* codeI63, cerebral infarction. The accuracy of diagnoses from claims data was verified in a previous study showing that the PPV and sensitivity of *ICD-10-CM* code I63 as a primary diagnosis of acute ischemic stroke were 92.7% and 99.4%, respectively [13]. We also analyzed the hazard ratio for comorbidities that were potentially linked to stroke: hypertension (I10–I13, I15), diabetes (E08–E11, E13), chronic kidney disease (CKD; N17–N19, I12, I13), chronic heart failure (I50), chronic obstructive pulmonary disease (J44, J60–70), hyperlipidemia (E78.0-E78.5), and atrial fibrillation (I48). The comorbidities were defined as the presence or absence of accompanying disease within one year before the index date of anemia. The national health insurance program (NHI) in Taiwan is mandatory for all citizens, and various medications and medical procedures were coded with unique code. In this study, six frequently prescribed drugs were included to investigate the efficacy of various anemia therapies for patients within six months after the hemoglobin measurement index date. These medications included iron (hydroxide-polymaltose complex, Yuanchou Chemical and Pharmaceutical Co., Ltd., Taiwan, NHI code AC46166100), ferric hydroxide sucrose complex (TCM Biotech international Corp. Taiwan, NHI code AC57884221), sodium ferrous citrate (Guang Heng Enterprise Co., Ltd. Taiwan, NHI code BC22097100), hydroxocobalamin (Shinlin Sinseng Pharmaceutical Co., Ltd. Taiwan, ACETATE, NHI code AC09754209), mecobalmin (Eisai Taiwan Inc., NHI code AC296301G0), folic acid (Johnson Chemical Pharmaceutical works Co., Ltd. Taiwan, NHI code AC346701G0), and blood transfusion (NHI code 94001C).

### 2.4. Statistical Analysis

Statistical analyses were conducted using the SAS statistical package (Version 9.4) and SPSS (version 28.0, SPSS Inc., Chicago, IL, USA) to examine the prevalence and clinical trends of anemia among the different age groups, sexes, and comorbidities. The categorical variables were assessed by applying a Chi-square test. The continuous variables were assessed by applying a t test. Furthermore, different predictors were used to estimate relative risks [14]. To examine the stroke risk associations with anemia, the deaths as competing risks of stroke were analyzed by using a Cox proportional cause-specific hazard model to calculate hazard ratios (HR), 95% confidence intervals (CIs), and two-sided *p* values. A two-sided *p* value of <0.05 was considered statistically significant. A multivariate Cox proportional cause-specific hazard regression model was adjusted for age, sex, and comorbidities. A proportional hazard assumption was evaluated by the Kolmogorov-type Supremum test; that was not violated.

## 3. Results

As shown in Figure 1, only 71,787 of the 454,424 patients who visited our facility qualified for the retrospective cohort research. The baseline characteristics of the case and control groups are summarized in Table 1. The mean Hb level was 14.2 ± 1.3 g/dL in the normal group and 10.7 ± 1.6 g/dL in the anemia group.

Of the 16,708 anemia patients, 6185 (37.02%) were men and 10,523 (62.98%) were women. The mean age of the case group was 59.1 ± 18.5 years, and that of the control group was 50.6 ± 16.3 years. The case group had a higher incidence of comorbidities, including hypertension (11.80% versus 20.40%, *p* < 0.001), diabetes (6.74% versus 14.77%, *p* < 0.001), CKD (0.90% versus 6.07%, *p* < 0.001), chronic heart failure (0.91% versus 2.67%, *p* < 0.001), chronic obstructive pulmonary disease (2.10% versus 2.96%, *p* < 0.001), and atrial fibrillation (0.49% versus 1.01%, *p* < 0.001), than did the control group.

Table 2 presents the anemia severity and subsequent cases of stroke. The patients with anemia were further divided into three subgroups according to anemia severity, determined on the basis of Hb levels by WHO criteria [12]. Thus, of the 16,708 patients with anemia, 9065 (54.25%) had mild anemia, 6532 (39.09%) had moderate anemia, and 1111 (6.65%) had severe anemia. During follow-up, a total of 447 anemia patients (2.68%, 447/16,708) and 744 controls (1.35%, 744/55,079) were diagnosed as having stroke. Moreover, there were 740 non-anemia patient deaths and 1229 anemia patient deaths throughout the 8-year follow-up period (1.34% and 7.63%, respectively).

We observed a positive association between the severity of anemia, determined based on Hb measurements, and the risk of stroke. Figure 2 illustrates the cumulative incidence of stroke in the three subgroups of anemia severity during the 8-year follow-up. A higher incidence of stroke events was noted in the patients with moderate anemia after their diagnosis during the 8-year follow-up (log-rank test, *p* < 0.001).

Table 3 illustrates the univariate and adjusted associations between the risk of stroke and the severity of anemia, sex, age, and comorbidities. The risk of stroke was higher in the case group than in the control group. In univariate regression analysis, we found moderate anemia (HR = 2.31; 95% CI, 1.97–2.71) had a significant increase in stroke risk compared to the non-anemia group. After adjusting, we found the risk of stroke was higher in the patients with moderate anemia (adj-HR, 1.20; 95% CI, 1.02–1.43, *p* = 0.032) than in the controls. The same results were obtained for gender and age by both univariate analysis (HR = 1.66, 95% CI = 1.48–1.87, *p* < 0.001; HR = 1.07, 95% CI = 1.07–1.08, *p* < 0.001, respectively) and adjusted HRs (adj-HR = 1.64, 95% CI = 1.46–1.85, *p* < 0.001; adj-HR = 1.07, 95% CI = 1.065–1.074, *p* < 0.001, respectively). Furthermore, the case group had a higher prevalence of comorbidities than did the control group. However, only the comorbidities diabetes mellitus and hyperlipidemia, by both univariate analysis (HR = 2.86, 95% CI = 2.50–3.28, *p* < 0.001; HR = 1.89, 95% CI = 1.54–2.31, *p* < 0.001, respectively) and adjusted HRs (adj-HR, 1.48; 95% CI, 1.27–1.71; *p* < 0.001), (adj-HR, 1.13; 95% CI, 0.91–1.39; *p* = 0.280), were associated with a higher risk of stroke in the case group compared to the control group.

## 4. Discussion

This retrospective study evaluated the prevalence and characteristics of anemia and the risk of stroke. The strength of this study is that it identified the association between anemia and the risk of stroke by using a hospital-based database, from which the laboratory data were retrieved to classify the severity of anemia. In contrast to previous studies, which have estimated the risk of stroke associated with anemia by using data from Taiwan’s NHI databases based on *ICD* codes and lacked conclusive laboratory Hb measurements [15,16], our study analyzed laboratory data and classified the patients into subgroups according to the severity of anemia to assess the associations between anemia severity and the risk of stroke. We also excluded patients with diseases that might interfere with our results, including those with a tendency of bleeding, other hemorrhagic disease, and persons who did not fulfill the predetermined criteria were also excluded. All participants in this study completed at least one Hb measurement, and persons who did not fulfill the predetermined criteria were then excluded. Our findings indicate that patients with moderate anemia showed an increased likelihood of stroke development.

In this retrospective analysis, there were more female anemic patients than male anemic patients. In the initial stage, the primary signs of mild anemia include fatigue, light skin, dizziness, debility, and headaches. Patients in the early stage of anemia or mild anemia may not seek medical care or consultations with physicians, particularly middle-aged men. Many male patients did not meet the criteria for hospital visits in 2 years. On the other hand, most women experience menopause at the age of 40–50 years; thus, some anemia symptoms, such as dizziness, fatigue, or paleness, may be overlooked or misdiagnosed as menopausal symptoms. Even when individuals visit a hospital or clinic, medical personnel tend to focus more on other maladies rather than anemia. However, if anemic condition is left untreated for a longer period, the consequences and complications can become more severe, causing shortness of breath, low blood pressure, arrhythmia, and even chronic heart failure. Results from this research demonstrated an increased risk of stroke occurrence in moderate anemia patients compared with the non-anemia control group. Additionally, the mortality rate in the severe anemia group was 12%, much higher than that of other patients with anemia in this study. Patients suffering from severe anemia might die from other illnesses caused by their feeble condition prior to having a stroke. As a result, the risk of stroke in the severe anemia group was observed to be lower than in the moderate anemia group.

According to statistical data from Taiwan’s Ministry of the Interior, the population aged >65 years increased from 11.15% in 2012 to 16.68% in 2021. In the past two decades, the average life expectancy also increased from 76.75 to 81.30 years. The Council for Economic Planning and Development estimated that Taiwan will become a super-aged society by as early as 2025; moreover, the population aged ≥65 years is expected to account for >20% of all individuals [17]. This accelerated speed of aging has become a burden to the healthcare system and society. In this study, there is an upward trend in the prevalence of anemia with age (from 6.72% in the 20–30 age range to over 15% in the elderly age groups; Table 1). Our results are consistent with the global prevalence of anemia, indicating that the trend of anemia burden increases with age [18,19,20]. We observed that the anemia prevalence peaked at 17.3% in the 71–80 age group and at 14.4% in the >80 age group. Anemia rates in the 71–80 age range in this study cohort were 4.2% (2990/71,787) and 3.4% (2411/71,787), respectively. In this study, the prevalence of anemia in people over 60 is approximately 11%, which is lower than it is in other Asian countries, such as Korea, where it is 13.8% for people over 65 [20]. Anemia is a common condition in older adults and can be caused by various factors such as poor nutrition, chronic diseases, medication, and healthcare. Taiwan has a relatively high standard of living, and the population has access to a variety of nutritious foods, which helps to prevent nutrient deficiencies, including iron deficiency. Moreover, Taiwan has a well-developed healthcare and medical insurance system. The Ministry of Health and Welfare also promotes and encourages all citizens above the age of 45 to participate in adult health checkup programs. These programs enable the early detection of diseases such as cancer and other chronic disease, as well as delivery of comprehensive healthcare prior to the disease worsening [17]. Therefore, all those factors may contribute to reduce the overall prevalence of anemia in the population. In elderly people, anemia has been reported to be associated with cardiovascular disease [21], stroke [6], dementia [22], frailty [23], and high morbidity as well as mortality [24]. Because of Taiwan’s NHI policy, however, anemia has rarely been listed as a primary condition in elderly people. According to the WHO recommendation, an anemia prevalence of >5% is considered to be of public health significance [12] and may require public health attention and intervention. The increased prevalence of anemia in the elderly should be considered an important public issue in Taiwan.

In this study, we also observed a higher prevalence of pre-existing comorbidities among the anemia group compared to the non-anemia population. The moderate to severe anemia patients had higher all-cause mortality compared to the non-anemia group; this trend was mentioned in previous studies [9,25]. Other unreported comorbidities may interfere with the association between anemia and stroke. Severe anemia might be corrected well, but mild to moderate anemia might become a chronic condition which eventually becomes associated with stroke. In this study, we observed that patients with severe anemia required blood transfusions more frequently than the group with moderate anemia and the control group. However, a study by Dr. Ren that was published in Nature Communications raises the possibility that blood transfusions might be advantageous to health even up to seven hours after a stroke in a mouse model. Their team discovered that replenishing 20% of the mouse’s blood was sufficient to significantly lessen brain damage [26]. However, there are few studies focusing on maintaining hemodynamic condition in severe anemia patients to prevent stroke. Therefore, more studies might help to clarify the benefit from blood transfusions on this issue in the future. Furthermore, the different therapeutic strategies may explain why severe anemia portends lower stroke risk than other anemia severities.

Studies assessing the association between anemia and comorbidities in the Taiwanese population are rare. Anemia, a direct consequence of decreases in Hb and red blood cell (RBC) levels in circulation, is a multifactorial condition; lack of iron, folate, and vitamin B12 are well-known causes of anemia. The most common type of anemia is iron deficiency anemia, which may account for as much as 50% of all explained anemia cases [27]. Other diseases such as diabetes, chronic infections, inflammation, and CKD also affect RBC proliferation, erythropoietin production, androgen secretion, and myelodysplasia [28]. Anemia is also positively associated with impaired renal function. Taiwan has one of the highest number of cases of CKD and end-stage renal disease in the world; CKD is the most frequent cause of anemia [8,20,21,29]. The severity of anemia is directly related to the degree of renal dysfunction. CKD causes reduction in erythropoietin synthesis, subsequently resulting in decreased cell proliferation. At least one-third of anemia patients aged >65 years have CKD or autoimmune diseases/chronic infection [30]. Patients with CKD are also at a significant risk for stroke, including the ischemic and hemorrhagic subtypes. The mechanisms linked to higher risk of stroke in CKD patients include alterations in cardiac output, platelet function, regional cerebral perfusion, accelerated systemic atherosclerosis, altered blood brain barrier, and disordered neurovascular coupling [31]. Additionally, Dr. Poznyak also identified the atherosclerosis-specific features in chronic kidney disease (CKD) in a recent study [32]. The major symptoms of anemia may range from mild fatigue to severe systemic illnesses. In addition, accumulating evidence indicates that anemia engenders outcomes such as increased stroke [9], heart failure [33], hospitalization [25], and mortality [34], all of which impose a severe burden on healthcare systems. Furthermore, anemia is associated with increased iron overload, increased chances of viral infection [35], and increased risks of myocardial infarction [36]. We also analyzed other known conventional risk factors, such as hyperlipidemia and atrial fibrillation, that affect the development of stroke; the hazard ratio was slightly different to other investigations [9]. Hyperlipidemia is an important risk factor for stroke [4,37]. Atrial fibrillation (AF) is a frequent cardiac rhythm disease associated with various significant negative health outcomes, such as heart failure and stroke. Particularly in women, atrial fibrillation is linked to an increased long-term risk of stroke, heart failure, and all-cause death [38,39]. Many investigations also revealed that anemia is a frequently observed comorbidity in patients with AF and is associated with cardiovascular, stroke, and gastrointestinal bleeding [40].

In medical practice, those experiencing moderate to severe anemia are more likely to receive medical attention than those with mild anemia. This means that patients with moderate to severe anemia with signs of illness symptoms would be given blood transfusions, iron supplements, and vitamin B12, while mild anemia would more likely be overlooked [41,42,43]. Regarding the management of anemic patients, blood transfusions are often seen as an effective way to increase hemoglobin levels and improve their overall health. In this study, we examined patients who received transfusions and pharmacological therapy within six months of the diagnosis index date. According to our results, patients with moderate and severe anemia received a greater proportion of blood transfusion than those with mild anemia (24.14%, 61.12% vs. 11.22%, Table 1). Blood transfusions can maintain in the body’s hemodynamics and alter the viscosity of the blood. Keeping the blood in balance in the body’s circulation and offering better care may be a strategy to prevent stroke. However, blood transfusion is influenced by a number of circumstances and the decision of the healthcare professionals. Patients who receive frequent transfusions may also be exposed to an increased risk of stroke. To ascertain the beneficial effects of anemia therapies such as transfusion and other medication on reducing the chance of stroke, further research must be conducted.

Despite its strengths, our study has some limitations that should be noted. First, the different types of anemia, such as iron deficiency anemia or folic acid anemia, were not correctly defined in this study. Second, we could not analyze data regarding lifestyles or socioeconomic status, such as smoking, alcohol habits, obesity, education, or financial condition. Third, in order to confirm the validity of the diagnosis for anemia, we only included the patients with one Hb measurement, which could cause a potential selection bias in a retrospective study. The medical service of our hospital serves a population of approximately 2.8 million in the center area of Taiwan, and more than 700,000 clinical visits are made each year. Finally, we did not retrieve clinical data on atherosclerosis, nutrition, pregnancy, or endogenous hormones, which might be predisposing factors for stroke and the retrospective data from the hospital might still miss a few stroke patients who were diagnosed in other hospitals or died at home.

## 5. Conclusions

This study assessed the association between anemia and the risk of stroke. The prevalence of anemia was found to increase with age. A high prevalence of anemia is expected to impose a major medical burden in countries becoming super-aged societies. In this study, the risk of stroke was found to be associated with age, regardless of sex. Our study reveals that moderate anemia should be considered an increased risk factor associated with stroke incidence, and monitoring anemia severity as well as other risk factors and biomarkers is crucial in clinical practice.

## Figures and Tables

**Figure 1 ijerph-20-03849-f001:**
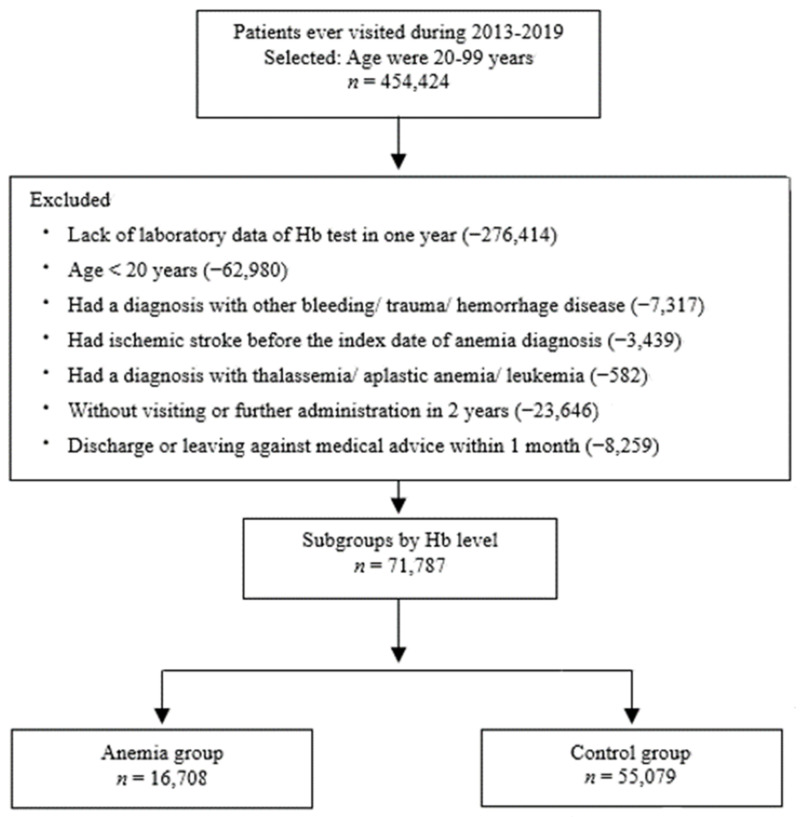
Flowchart of the patient enrollment process in this study. A total of 71,787 patients were included in this study. In total, 16,708 patients were subgrouped into the anemia group and 55,079 patients were subgrouped into the normal group.

**Figure 2 ijerph-20-03849-f002:**
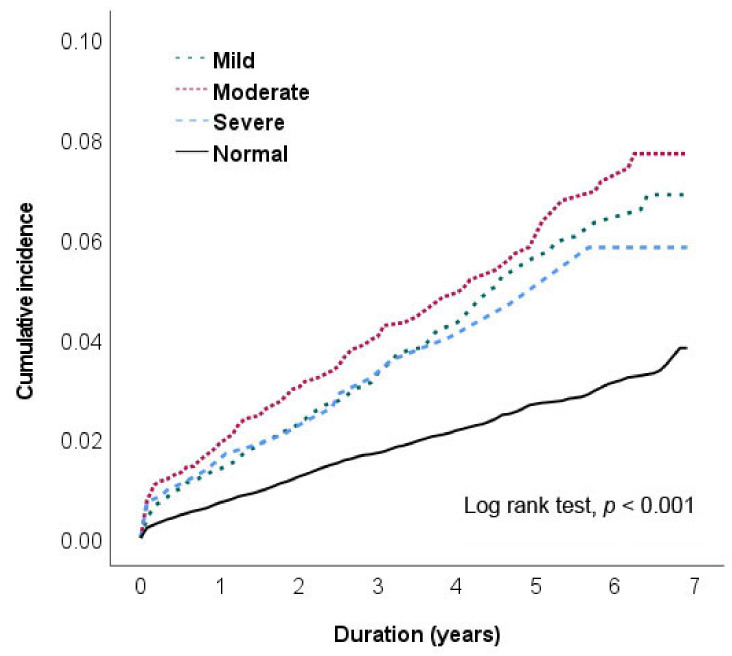
Cumulative risk of stroke based on the anemia severity during the 8-year follow-up: mild anemia (green curve), moderate anemia (red curve), severe anemia (blue curve), and non-anemia (controls; black curve).

**Table 1 ijerph-20-03849-t001:** Baseline characteristics of study cohort.

Characteristic *n* = 71,787	Normal *n* = 55,079 (76.73%)		All Anemia *n* = 16,708 (23.27%)		*p*	Mild Anemia		Moderate Anemia		Severe Anemia	
						*n* = 9065 (53.99%)		*n* = 6532 (39.27%)		*n* = 1111 (6.75%)	
Hb (g/dL)	14.2 ± 1.3		10.7 ± 1.6		<0.001	11.8 ± 0.5		9.9 ± 0.8		6.7 ± 1.1	
Gender											
Male	26,184	47.54%	6185	37.02%		3933	43.39%	1880	28.78%	372	33.48%
Female	28,895	52.46%	10, 523	62.98%		5132	56.61%	4652	71.22%	739	66.52%
Age (years)	50.6 ± 16.3		59.1 ± 18.5		<0.001	58.9 ± 18.5		59.7 ± 18.6		57.9 ± 17.5	
20–30	7298	13.25%	1123	6.72%	<0.001	704	8.1%	705	7.78%	369	5.65%
31–40	9139	16.59%	1961	11.74%		1069	12.3%	1077	11.88%	756	11.57%
41–50	10,423	18.92%	2883	17.26%		1287	14.9%	1316	14.52%	1275	19.52%
51–60	12,475	22.65%	2586	15.48%		1451	16.8%	1489	16.43%	919	14.07%
61–70	9168	16.65%	2754	16.48%		1512	17.5%	1610	17.76%	999	15.29%
71–80	4732	8.59%	2990	17.90%		1536	17.7%	1671	18.43%	1147	17.56%
81 above	1844	3.35%	2411	14.43%		1100	12.7%	1197	13.20%	1067	16.33%
Comorbidities											
Hypertension	6497	11.80%	3408	20.40%	<0.001	1797	19.82%	1400	21.43%	211	18.99%
Diabetes	3713	6.74%	2467	14.77%	<0.001	1254	13.83%	1067	16.33%	146	13.14%
Chronic kidney disease	496	0.90%	1015	6.07%	<0.001	300	3.31%	580	8.88%	135	12.15%
Chronic heart failure disease	500	0.91%	446	2.67%	<0.001	206	2.27%	208	3.18%	32	2.88%
Chronic obstructive pulmonary disease	1159	2.10%	495	2.96%	<0.001	293	3.23%	179	2.74%	23	2.07%
Hyperlipidemia	2177	3.95%	643	3.85%	0.544	395	4.36%	222	3.40%	26	2.34%
Atrial fibrillation	270	0.49%	169	1.01%	<0.001	86	0.95%	73	1.12%	10	0.90%
Treatment											
Blood transfusion	1795	3.26%	3273	19.59%	<0.001	1017	11.22%	1577	24.14%	679	61.12%
Iron therapy	72	0.13%	662	3.96%	<0.001	67	0.74%	388	5.94%	207	18.63%
Folic acid supplement	344	0.62%	469	2.81%	<0.001	135	1.49%	226	3.46%	108	9.72%
Vitamin B12 supplement	68	0.12%	94	0.56%	<0.001	37	0.41%	38	0.58%	19	1.71%

**Table 2 ijerph-20-03849-t002:** Severity of anemia classified according to WHO criteria and subsequent stroke events.

Severity of Anemia	WHO Criteria (Hb, g/dL)		Study Cohort *n* = 71,787		Stroke Events *n* = 1191		Death Events *n* = 1969		Average Follow Up
	Male	Female	*n*	%	*n*	%	*n*	%	Years
Normal	>13.0	>12.0	55,079	―	744	1.35	740	1.34	2.32 ± 2.04
Anemia			16,708		447	2.68	1229	7.63	
Mild	11.0–12.9	11.0–11.9	9065	54.25	229	2.53	496	5.47	2.09 ± 1.95
Moderate	8.0–10.9	8.0–10.9	6532	39.09	193	2.95	599	9.17	1.98 ± 1.95
Severe	<8.0	<8.0	1111	6.65	25	2.25	134	12.06	1.95 ± 1.94

**Table 3 ijerph-20-03849-t003:** Risk associations between stroke and anemia, sex, age, and comorbidities.

Predictors	HR (95% CI)			
	Univariate	*p*	Adjusted	*p*
Severity of anemia				
Normal	―		―	
Mild	1.96 (1.69–2.27)	<0.001	0.98 (0.84–1.15)	0.795
Moderate	2.31 (1.97–2.71)	<0.001	1.20 (1.02–1.43)	0.032
Severe	1.73 (1.16–2.58)	0.007	0.99 (0.66–1.48)	0.943
Gender (male)	1.66 (1.48–1.87)	<0.001	1.64 (1.46–1.85)	<0.001
Age (years)	1.07 (1.07–1.08)	<0.001	1.07 (1.065–1.074)	<0.001
Comorbidity				
Hypertension	3.11 (2.76–3.50)	<0.001	1.26 (1.10–1.45)	0.001
Diabetes mellitus	2.86 (2.50–3.28)	<0.001	1.48 (1.27–1.71)	<0.001
Chronic kidney disease	2.65 (2.10–3.35)	<0.001	1.02 (0.80–1.31)	0.869
Chronic heart failure disease	3.18 (2.41–4.19)	<0.001	1.00 (0.74–1.35)	0.988
Chronic obstructive pulmonary disease	1.88 (144–2.46)	<0.001	0.76 (0.58–1.00)	0.053
Hyperlipidemia	1.89 (1.54–2.31)	<0.001	1.13 (0.91–1.39)	0.280
Atrial fibrillation	5.50 (3.98–7.59)	<0.001	1.84 (1.31–2.60)	<0.001

HR: hazard ratio; CI: confidence interval. *p* < 0.05 was considered statistically significant.

## Data Availability

The datasets generated and analysed in this study are not publicly available due to Tzu Chi Hospital regulations.

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
