# Peer review of "Association between Anemia Severity and Ischemic Stroke Incidence: A Retrospective Cohort Study"

_ijerph, 2023, doi:10.3390/ijerph20053849_

Round 1

Reviewer 1 Report (New Reviewer)

Authors assessed the association of different grades of anemia severity  with the risk of stroke in a cohort of 69,538 patients with a follow up of 8 years. In a multivariate analysis, after adjusting for sex, age and other comorbidities they show that moderate anemia is associated with the risk of stroke.

 Major Points:

-          Codes I65 and I66 refer to stenosis of cerebral and precerebral arteries, not resulting in cerebral infarction. Please remove from study.

-          Lines 219-221 mention that your prevalence is very low compared to other studies (3.6% vs 15-30%), but line 247 data of prevalence between 7 to 15%, more close to global data (15-30%). Mention why you provide different prevalences, and which one is better to consider.

-          Lines 248-249: Your data is not consistent with ref.18 and 19 data. Table 1 shows that in your cohort the peak is at 71-80 years (17.3%), being lower at over 81 (13.9%). Please discuss this difference between your cohort and other similar studies.

-          Lines 279-281: If frequent blood transfusions could be associated to stroke in people with anemia, why severe anemia is not assocated to stroke? Please, focus discussion only on demonstrated results, that is association fo Moderate anemia to stroke.

-          Line 310: The study identified the association of moderate anemia with the risk of stroke.

-          The authors should discuss why in their cohort the prevalence is higher in women and in other cohorts (Ref.18 and 19 for example).

-          Include Death due to Cardiovascular events in table 2.

-          Show other CVD in anemia population.

-          Show table with all the characteristics (Gender, Age, comorbidities, Treatments and anemia (normal mild, moderate and severe) according to Stroke presence

c   Discuss why only Moderate anemia and not Mild and severe are associated with stroke risk

Minor Points:

-          Is possible to provide data after 4 years follow up?

-          Lines 38-40: “Many well-known stroke risk factors, including hypertension, hyperlipidemia, diabetes mellitus, tobacco use, obesity, aging, and physical activity [1, 4, 5].”

Please rephrase

-          Ref 10 shows more recent data on global prevalence of anemia than reference 9, and data differs between them

-          Lines 51-53: “Although some studies showed anemia may increase risk of stroke. However, American Stroke Association (ASA) do not recognize anemia as a major risk factor of stroke in the updated stroke guideline [11].”

Please rephrase

-          Lines 134-137: Please rephrase

-          Line 142: “pervious”

-          Line 162: “there were 454,424 patients came to our hospital”

Please rephrase

-          Lines 181-182: “Furthermore, during the 8-year follow-up, 1,136 patients with anemia (7.1%) and the number of death in non-anemia patient was 663 (1.2%).”

Please rephrase.

-          Lines 217-219: “In this 217 study, all patients underwent at least one Hb measurements and other excluded criteria 218 and comorbidities aforementioned were excluded”

Please rephrase

-          Line 235: “Positive”

-          Lines 264-268: Provide references supporting your affirmations.

-          Lines 294-297: CKD is also involved in atherosclerosis development and its assocated CVE (includding stroke). Mention it.

-          Line 303: “atrial fibrillation that affect the development of stroke”

Please rephrase.

-          Line 304: “Hyperlipidemia is an important risk factor for stroke”

Provide reference

-          Line 307: “Many investigations also revealed that anemia with AF is associated..” rephrase to something like “Many investigations also revealed that anemia is a frequently observed comorbidity in patients with AF and is associated…”  

Author Response

Dear Reviewer:

I hereby submit my revised manuscript (manuscript number ijerph-2171456) entitled entitled "Association between Anemia Severity and Ischemic Stroke Incidence: A Retrospective Cohort Study" to be considered for publication in Special Issue "2nd Edition: Frontiers in Health Care for Older Adults" as a full paper. This manuscript has been carefully revised according to the comments of the Reviewers. A detailed itemized response to Reviewers’ comments as the attachment.

Reviewer 2 Report (New Reviewer)

Thank you for giving an opportunity to review this manuscript. Authors conducted a cohort study to evaluate the risk analysis of anemia on development of stroke using data from a single hospital. Authors concluded that there is a high risk of stroke in patients with anemia at long-term follow-up. My comments are:

Major comment

1. The stroke outcome was defined as at least two OPD visits or related hospitalizations with ICD-10 codes I60-I63, I65, I66-68, G45-G46 (except G453, G454, I673, I6783). I65 and I66 are diagnostic codes for cerebrovascular stenosis. Why are they included in the outcome variable? If there is a related reference, citation and description will be required.

2. It is thought that comorbidities that can be risk factors for stroke have been identified with each diagnosis code. Have you confirmed that these diseases have been treated in OPD or hospitalized more than 2 times?

 3. Please indicate whether the treatment method was confirmed through chart review or claim code in the method section.

4. Isn't the p-value statistically meaningless in the cox proportional regression analysis because the ratio of comorbidities is lower than that of other study subjects?

 5. In the discussion, it is mentioned that patients with severe anemia receive treatment, but patients with mild anemia tend not to receive treatment. Did the analysis include the effect of the method (transfusion or medication) or presence of treatment on the risk of stroke?

6. One of the reasons why patients with severe anemia appear to have a low risk of stroke is that they die from other diseases before stroke onset due to poor general condition. In this study, the mortality rate of the severe anemia group was 11.7%, which was higher than that of the other anemia groups.

 7. As the authors analyzed the outcomes including both ischemic and hemorrhagic strokes, a mechanism for the risk of anemia for the development of each stroke should be added to the discussion section.

Minor comment

1. Figure 2 differs from the illustration below. Is the green curve for moderate anemia and the blue curve for severe anemia?

2. In the statistical analysis, the stroke risk was evaluated over time, so modify it to a time-dependent cox regression analysis.

Author Response

Dear Reviewer:

I hereby submit my revised manuscript (manuscript number ijerph-2171456) entitled entitled "Association between Anemia Severity and Ischemic Stroke Incidence: A Retrospective Cohort Study" to be considered for publication in Special Issue "2nd Edition: Frontiers in Health Care for Older Adults" as a full paper. This manuscript has been carefully revised according to the comments of the Reviewers. A detailed itemized response to Reviewers’ comments has been uploaded to the journal manuscript website.

Round 2

Reviewer 1 Report (New Reviewer)

The authors have adequately responded all comments, which has significantly improved the article's quality.

I believe that the article can be published in IJERPH as it now is.

I only have two minor comments: 

Line 180. The case group hasn’t had higher prevalence of hyperlipidemia.

Lines 189-190. “there were 740 non-anemia patient deaths and 1,229 anemia patient deaths throughout the 8-year follow-up period (7.63% and 1.34%, respectively).” % values are flipped.

Author Response

We thank the Reviewers for careful reading of our manuscript and providing insightful comments. The manuscript has been revised according to the suggestions and comments of the reviewers. All minor annotations have been corrected to this manuscript and indicated by the red colored text.

Reviewer 2 Report (New Reviewer)

The authors corrected the manuscript appropriately in response to most reviewers' comments. The content of the manuscript has been greatly developed compared to the previous one.

Correct only 2 sentence errors in the manuscript.

The order of Line 190, 7.63% and 1.34% is reversed.

Line 274, received comprehensive health17care prior to disease worsening. Is 17 the reference number?

Author Response

We thank the Reviewers for careful reading of our manuscript and providing insightful comments. The manuscript has been revised according to the suggestions and comments of the reviewers. All minor annotations have been corrected to this manuscript and indicated by the red colored text.

This manuscript is a resubmission of an earlier submission. The following is a list of the peer review reports and author responses from that submission.

Round 1

Reviewer 1 Report

Chen et al present a retrospective cohort study examining the relationship between stroke and anemia in a large cohort (a total of 106,478 patients). The main limitation of this study is the lack of any multivariate analysis adjusting for other stroke risk factors in order to clarify the etiologic relationship between stroke and anemia. Taking into account that the groups were matched a conditional logistic regression should be performed. Authors should revise their statistical analysis and present the multivariate analysis adjusting for (at least) Hypertension, CKD, DM, Chronic obstructive pulmonary disease, HF.

Minor comments:
Abstract:
1. Authors should rephrase the sentence "Stroke causes permanent brain damage and constant complications and is associated with anemia." Anemia is not the first risk factor of stroke that comes in a neurologist's mind.
2. Authors should rephrase the sentence "Accordingly, mild anemia is a risk factor for stroke, and routine health checks and earlier intervention can help prevent anemia, especially in middle-aged and elderly individuals". It implies that if we prevent anemia we may prevent stroke which is not what this study examined.
Results:
1. Authors should clarify whether all patients completed the reported 8-year follow-up period. Otherwise they should define the number of patients lost at follow-up.
2. At figure 3 cerebrovascular disease appears for the first time as an examined comorbidity while previous stroke was an exclusion criterion. Please clarify.
Discussion:
1. An important limitation is the retrospective nature of this study that should be discussed.
2. Authors should discuss the fact that they did not assess hyperlipidemia and AF as a stroke risk factor in study's population.
3. Considering the higher prevalence of comorbidities in anemia group which may explain the higher incidence of stroke in this group (as clearly presented at figure 3) all the statements about the relationship between anemia and stroke should be rephrased.This should be also discussed as a limitation.

Author Response

We thank the Reviewers for careful reading of our manuscript and providing insightful comments. The manuscript has been revised according to the suggestions and comments of the reviewers. Additionally, all the changes to this manuscript have been marked by red-colored texts. Following is the list of our responses to the comments of Reviewers:

Reviewer 2 Report

Chen et al. report a hospital database analysis on the association between anemia and stroke. Most of the study is presented clearly and the methods, statistical analysis and the conclusions seem appropriate.

I have some minor comments regarding information that is missing and some sentences that are unclear.

- Abstract: The authors should add that the HR for stroke are calculated compared to no anemia.

- Methods: The authors should give some information on the Taichung Tzu-Chi Hospital. How large is the catchment area and how plausible is it that persons living in this area are admitted to this hospital if the suffer are stroke. It seems that the diagnosis of stroke is based on persons admitted at least twice to the same hospital. The authors should explain in the methods ((2.4. outcome) and / or the discussion how many stroke cases might have been missed, i.e. how reliable the outcome “stroke” is. Do the authors have information on patients which may have died or moved out of the area during the 87 years? The authors should discuss possible bias in the limitations section of the discussion

- lines 77-79, “Patients who were aged <20 years and had a diagnosis […]”: This sentence seems a bit unclear. It seems that patients younger than 20 years were excluded anyway, independent of stroke or anemia? Otherwise, this would mean that persons younger than 20 years without stroke and without anemia were included in the study.

- line 86 Typo: 8.0 instead of 80 for severe anemia

- lines 146, “outcomes were assessed at baseline”: I suppose “stroke” is the outcome and that baseline was the first Hb blood level measurement. Thus I would think that the outcome was measured during the 2 or 8 years?? Please clarify.

- line 152 -155: Most comorbidities are risk factors for stroke. At which time point were these comorbidities diagnosed, i.e. at the index date, at 2 years, at 8 years?

- line 242: “anemia can be more effective”: please change the wording, this is unclear.

- Discussion: The authors should discuss that anemia might be treated in the 2, respective 8, years since the index date. Furthermore, the analysis suggest an inverse relationship between stroke risk and anemia. However, this might be due to other factors such as the treatment of severe anemia which is highly probably in hospital admitted patients and less treatment of mild anemia, as discussed in line 243 -251. The authors should clarify these paragraphs, especially those on severe anemia.

- Discussion: The authors should discussion different sources of bias. E.g. Hb levels were only measured in 1/3 of patients. It is possible that Hb levels were measured in patients with specific conditions, leading to a selection bias. Furthermore, patients with stroke were admitted twice, whereas those without stroke might not have been admitted, died, lost to follow up etc..

Author Response

(The authors gave the same response as above.)

Round 2

Reviewer 1 Report

Authors added the multivariate analysis as requested and significantly changed their manuscript. However, they still use the univariate analysis as the main one to support their original hypothesis. For example, at figure 3 forest plot represents univariate analysis and at the abstract they state that "Mild and moderate anemia were associated with a relatively high risk of stroke (mild anemia: HR, 2.07; 95% CI, 1.51–26 2.85, P <0.001". However, at the multivariate analysis p for mild anemia is non significant. We could not assume that "the proportions of missed strokes were theoretically even in anemia group and control group". As stated before taking into account that the groups were matched a conditional logistic regression should be performed rather than multivariate cox proportional hazard regression model . Overall, the scientific soundness of the relationship between stroke and anemia is very low as presented in this manuscript.